Antagonistic co-contraction can minimize muscular effort in systems with uncertainty

Koelewijn Anne D. 1 2 anne.koelewijn@fau.de
http://orcid.org/0000-0002-3791-3749 Van Den Bogert Antonie J. 2
1 Machine Learning and Data Analytics (MaD) Lab, Department Artificial Intelligence in Biomedical Engineering, Friedrich-Alexander Universität Erlangen-Nürnberg , Erlangen , Germany
2 Parker-Hannifin Laboratory for Human Motion and Control, Department of Mechanical Engineering, Cleveland State University , Cleveland, Ohio , United States
van Dieën Jaap
Electronic publication date: 2022 Apr 7
Publication date: 2022
Volume: 10
Electronic Location ID: e13085
Received 2021 Oct 13; Accepted 2022 Feb 17
Copyright: © 2022 Koelewijn and Van Den Bogert
Copyright year: 2022
Copyright holder: Koelewijn and Van Den Bogert
License: This is an open access article distributed under the terms of the Creative Commons Attribution License, which permits unrestricted use, distribution, reproduction and adaptation in any medium and for any purpose provided that it is properly attributed. For attribution, the original author(s), title, publication source (PeerJ) and either DOI or URL of the article must be cited.
License URL: https://creativecommons.org/licenses/by/4.0/

Keywords: Antagonistic co-contraction, Co-activation, Optimal control, Muscle mechanics, SCONE, Muscular effort, Effort minimization

Funding: National Science Foundation 1344954 Parker-Hannifin Corporation Deutsche Forschungsgemeinschaft and Friedrich-Alexander-Universität Erlangen-Nürnberg This research was supported by the National Science Foundation (grant number 1344954), by a Graduate Scholarship from the Parker-Hannifin Corporation, and by a faculty endowment from Adidas. We received financial support from Deutsche Forschungsgemeinschaft and Friedrich-Alexander-Universität Erlangen-Nürnberg within the funding programme “Open Access Publication Funding”. The funders had no role in study design, data collection and analysis, decision to publish, or preparation of the manuscript.

==============================
Muscular co-contraction of antagonistic muscle pairs is often observed in human movement, but it is considered inefficient and it can currently not be predicted in simulations where muscular effort or metabolic energy are minimized. Here, we investigated the relationship between minimizing effort and muscular co-contraction in systems with random uncertainty to see if muscular co-contraction can minimize effort in such system. We also investigated the effect of time delay in the muscle, by varying the time delay in the neural control as well as the activation time constant. We solved optimal control problems for a one-degree-of-freedom pendulum actuated by two identical antagonistic muscles, using forward shooting, to find controller parameters that minimized muscular effort while the pendulum remained upright in the presence of noise added to the moment at the base of the pendulum. We compared a controller with and without feedforward control. Task precision was defined by bounding the root mean square deviation from the upright position, while different perturbation levels defined task difficulty. We found that effort was minimized when the feedforward control was nonzero, even when feedforward control was not necessary to perform the task, which indicates that co-contraction can minimize effort in systems with uncertainty. We also found that the optimal level of co-contraction increased with time delay, both when the activation time constant was increased and when neural time delay was added. Furthermore, we found that for controllers with a neural time delay, a different trajectory was optimal for a controller with feedforward control than for one without, which indicates that simulation trajectories are dependent on the controller architecture. Future movement predictions should therefore account for uncertainty in dynamics and control, and carefully choose the controller architecture. The ability of models to predict co-contraction from effort or energy minimization has important clinical and sports applications. If co-contraction is undesirable, one should aim to remove the cause of co-contraction rather than the co-contraction itself.

Introduction

Understanding human movement is one of the key goals in biomechanics research. During the last 60+ years, the energy optimality of movement has been shown in different experimental and simulation work. Using experiments, Ralston showed that people choose their walking speed to minimize the metabolic energy expenditure per distance travelled in 1958 (Ralston, 1958). Since then, the minimization of energy expenditure in movement has been confirmed for different other parameters, such as step frequency (Zarrugh, Todd & Ralston, 1974), step width (Donelan et al., 2004) and vertical movement of the center of mass (Ortega & Farley, 2005; Gordon, Ferris & Kuo, 2009). Furthermore, walking and running emerge as energy-optimal gaits at their respective speeds when optimal simulations are created with simple walking models (Srinivasan & Ruina, 2006). Static optimization has revealed that muscle forces can be explained by minimizing an objective related to effort (Crowninshield & Brand, 1981). Simulations with complex musculoskeletal models, where an objective related to energy or effort is minimized, have also revealed a motion that looks very similar to walking (Ackermann & Van den Bogert, 2010; Koelewijn, Dorschky & Van den Bogert, 2018). Furthermore, energy or effort minimization makes sense from an evolutionary perspective as well (Wall-Scheffler, 2012).

However, certain human behaviours seem to contradict the notion that movements minimize energy or effort, a prime example being antagonistic co-contraction of muscles. Antagonistic co-contraction is the activation of both agonist muscles, which support a movement, and antagonist muscles, which oppose a movement, around a joint. It increases the instantaneous muscle stiffness due to the nonlinear mechanical properties of the muscle, and consequently prevents movement. Co-contraction does not produce external forces or work while it requires effort and metabolic energy (Hogan, 1984), and therefore co-contraction is often described as inefficient (Falconer & Winter, 1985; Winter, 2005). The benefit of co-contraction in human movement has been described as an increase in joint stiffness and stability (Hogan, 1984; Hirokawa et al., 1991; Jiang & Mirka, 2007; Selen, Beek & van Dieën, 2005), a reduction of stress in the joint ligaments (Baratta et al., 1988), and lower tibial shear force (Baratta et al., 1988).

In this paper, we aim to show that, surprisingly, co-contraction minimizes energy or effort in practice, due to noise in the dynamics. Noise can be caused by external uncertainty, such as due to wind, but noise is also present internally in sensory and motor neurons (Bays & Wolpert, 2007). As a result, human control should constantly correct any deviations caused by noise. These corrections can be done pro-actively or re-actively, where a pro-active approach would be to use co-contraction, and the related increase in joint stiffness, to prevent movements, while a re-active approach would be to correct for disturbances through feedback. Human feedback control is known to have time delay between the sensation and the correction (Carpenter, Allum & Honegger, 1999), and Hogan (1984) has also related the need for co-contraction to this time delay, since it makes feedback control less efficient. Therefore, we also aim to investigate how the optimality of co-contraction depends on the time delay in the system.

To investigate the optimality of co-contraction, and its connection to time delay, we can use tools from stochastic optimal control. Based on the knowledge that humans minimize energy or effort, researchers have studied human movement using tools from optimal control (He, Levine & Loeb, 1989; Park, Horak & Kuo, 2004; Ackermann & Van den Bogert, 2010). Then, the goal is to find the optimal input, which could be described by a control law (He, Levine & Loeb, 1989; Park, Horak & Kuo, 2004), or parameterized over time (Ackermann & Van den Bogert, 2010), that minimizes an objective related to energy or effort, while performing the desired task. However, commonly, the dynamics are described using a deterministic model, which does not account for the internal or external noise (Anderson & Pandy, 2001; Miller, Brandon & Deluzio, 2013; Ackermann & Van den Bogert, 2010; Koelewijn, Dorschky & Van den Bogert, 2018). Also, human systems are often linearized (He, Levine & Loeb, 1989; Park, Horak & Kuo, 2004), and e.g. linear quadratic Gaussian control (Todorov, 2005) is used to find an optimal controller for the linear system. Due to the certainty equivalence principle, the optimal controller and trajectory for a linear deterministic system is the same as for its equivalent stochastic system (Anderson & Moore, 1989), so with this approach noise does not need to be added to find the optimal solution. As a consequence, co-contraction would never be optimal, since a solution without co-contraction always requires less energy or effort without noise than one with co-contraction. However, human dynamics, both of the skeleton and the muscles, are nonlinear, while the internal noise is signal dependent and thus nonlinear as well, meaning that the certainty equivalence principle does not hold for human movements. Instead, to account for the noise, we should use stochastic optimal control while allowing for the system to be nonlinear.

However, the solution to stochastic optimal control problems can only be estimated using time-consuming approaches. When solving a stochastic optimal control approach for nonlinear systems, the stochastic Hamilton–Jacobi–Bellmann equations should be solved. However, these equations become intractable for high dimensions (Kappen, 2005). Therefore, several methods have been developed that estimate the solution of optimal control problems for nonlinear systems. The most commonly used approach is the Monte Carlo method (Tiesler et al., 2012; Sandu, Sandu & Ahmadian, 2006a). In this method, a large number of forward simulations are performed, where the uncertain variables are sampled from their distribution. Using the solution of each of the forward simulations, a distribution of the outcome variables is provided. This method requires a large number of simulations to obtain an accurate solution (Sandu, Sandu & Ahmadian, 2006a). Another approach is based on the theory of generalized polynomial chaos, which states that a second order stochastic process can be approximated by a combination of stochastic basis functions. When combined with a collocation method, this approach can solve optimal control problems faster than a Monte Carlo method (Tiesler et al., 2012; Sandu, Sandu & Ahmadian, 2006a, 2006b). We have recently developed such an approach to solve stochastic optimal control problems for human gait based on sampling and direct collocation (Koelewijn & van den Bogert, 2020). However, this method still does not work well for dynamics models with muscles, where the relationship between control and rigid body dynamics is nonlinear.

Instead of using collocation, forward shooting could be an appropriate alternative to investigate our hypothesis. Historically, optimal control problems of gait were solved with forward shooting, but these problems required many computer hours to solve, and often gait cycles were not periodic (Anderson & Pandy, 2001; Miller, Brandon & Deluzio, 2013). Direct collocation reduced the optimization time to less than 1 h for three-dimensional gait simulations (Falisse et al., 2019; Nitschke et al., 2020) and thereby greatly enhanced the possibilities for gait simulations. Recently though, advances have also been achieved using forward shooting, especially in combination with reflex models (Ong et al., 2019; Koelewijn & Ijspeert, 2020), and nowadays this method, e.g. using the software SCONE (Geijtenbeek, 2019), can also efficiently solve optimal control problems with a predefined controller structure and a relatively smaller number of optimization variables.

To test our hypothesis that co-contraction is optimal in a system with noise, and to investigate the relation between co-contraction and time delay, we should therefore use a nonlinear system with uncertainty. We will solve optimal control problems on the simplest possible problem where co-contraction can occur: to find the optimal muscle controls for a one degree of freedom pendulum, controlled by two muscles at the base, to remain upright while noise is applied to the pendulum base. These problems will be solved using forward shooting in SCONE. Similar to Crowninshield & Brand (1981), we use effort minimization as an objective. We use this problem to investigate if a control strategy with more co-contraction requires less effort than a control strategy without co-contraction for certain tasks in systems with uncertainty, even when a strategy without co-contraction is possible. To do so, we will first visualize the effort landscape over different co-contraction levels for an example problem to investigate if a control strategy with co-contraction requires less effort than a strategy without co-contraction. Then, we will perform a comparison of different tasks with varying precision and difficulty to investigate in which conditions co-contraction is optimal. Next, we will examine the relationship between co-contraction and time delay to investigate if co-contraction is indeed required due to time delay in human feedback control, as suggested by Hogan (1984). Since time delay is present both in the muscle and as a neural time delay between a sensory stimulus and an action, we will investigate both the effect of increased neural time delay and the effect of a longer activation time.

Methods

We developed a one degree of freedom pendulum model, operated via two muscle-tendon units (MTUs) in OpenSim (Seth et al., 2018) (Fig. 1). The pendulum was modelled as a point mass of 1 kg, located 50 cm from a revolute joint, which connected the pendulum to the ground. We set the default value of its degree of freedom to π, or the upright position, which we defined as the angle θ = 0. Gravity was pointing downwards. We attached identical MTUs on both sides of the revolute joint to operate the pendulum. These MTUs were attached to the ground with a ±10 cm offset, and attached to the pendulum at 10 cm above the joint. The MTUs were modelled as “Thelen2003Muscle” (Thelen, 2003) with identical parameters (Table 1). The optimal fiber length and tendon slack length were chosen such that the muscle was approximately slack in the upright position.

Figure 1 Pendulum used in this study.

The pendulum has one rotational degree of freedom at the base, and is operated by two identical muscle-tendon units (MTUs). The MTUs are modeled as “Thelen2003Muscle” (Thelen, 2003).

Table 1 Default muscle model parameters.

Parameter	Value and unit	
Maximum isometric force	Fmax = 2,000 N	
Maximum shortening velocity	vCE(max) = 10 lCE(opt)/s	
Maximum force during lengthening	gmax = 1.8 Fmax	
Optimal fiber length	lCE(opt) = 7 cm	
Activation time constant	Tact = 10 ms	
Deactivation time constant	Tdeact = 40 ms	
Tendon slack length	lSEE,slack = 8 cm	
Pennation angle at optimal	ϕopt = 0.2 rad	
Tendon strain at isometric force	eSEE,iso = 0.033	
Passive muscle strain at isometric force	ePEE,iso = 0.6	
Active force-length shape factor	Kact = 0.5	
Passive force-length shape factor	Kpas = 4	
Force-velocity shape factor	Af = 0.3	

Controller model

We designed two controllers to determine the MTU input from the pendulum angle and angular velocity. To compare the effect of co-contraction, one controller had feedforward control and feedback control, while the other only had feedback control. In the first controller, we created a constant feedforward stimulation, which represents co-contraction for a static task. We created feedback control using proportional-derivative control, which penalized any deviation from the upright position and any angular velocity. Since the problem is symmetric, the feedforward control was the same for both MTUs, while the feedback control had an opposite sign between the MTUs. Therefore, the full input for muscle i, ui, was calculated as follows for the controller with feedforward and feedback control:

(1) ui(t)=u0±(KPθ(t−Δt)+KDω(t−Δt)),

where u0 denotes the feedforward control, KP the position feedback gain, KD the derivative feedback gain, ω the angular velocity of the pendulum, and Δt the neural time delay.

The second controller was exactly the same, but we set the feedforward control to 0.01, the minimum muscle activation that is allowed in OpenSim and SCONE, such that only feedback control was used:

(2) ui(t)=0.01±(KPθ(t−Δt)+KDω(t−Δt)).

Optimal control problem and simulations

We solved optimal control problems to optimize controller parameters for different tasks. The goal was to find the control parameters that minimized muscular effort during a 20 s simulation while remaining close to the upright position position, under the influence of perturbations added to the base of the pendulum, since we aim to show that co-contraction minimizes energy or effort in systems with uncertainty. We defined the precision of the tasks using a maximum root mean square (RMS) deviation from the upright position over the full simulation. We chose this task description instead of bounding the maximum deviation to ensure that co-contraction was not required due to one large perturbation that could not be overcome by the muscles otherwise. We also defined a fall, which ended the simulation, when the joint angle deviated more than 1 rad from the upright position. All simulations started from the upright position and zero angular velocity. This yields the following optimization description:

(3) For dynamic systemx˙=f(x(t),u(t))

(4) with initial conditionsθ(0)=0,ω(0)=0,

(5) minimizeu0,KP,KDJ(u(t))=12T∫t=0Tu1(t)2+u2(t)2dt

(6) subject toθRMS=1T∫t=0Tθ(t)2dt≤θRMS,max

(7) T≥20,

where f(x(t), u(t)) describe the dynamics, derived with OpenSim (Seth et al., 2018), T the duration of the simulation, θRMS the RMS deviation of the angle, and θRMS,max the maximum RMS deviation, or the desired task precision.

We used SCONE (Geijtenbeek, 2019) to solve these optimal control problems via forward shooting. Since SCONE does not allow for constraints, we constructed an objective, JSCONE, to ensure that effort was minimized, while our constraints were met. To do so, we added two objective terms to the objective of minimizing effort, such that the RMS deviation was penalized once it was larger than maximum RMS deviation, and another penalty was added when a simulation finished in less than 20 s.

These objectives were added in such a way that they were equal to 0 when the constraints were met, and nonzero otherwise, such that, for valid solutions, the objective is purely effort minimization. The ratio between the objectives weights was chosen such that the constraints were met in a sensible order. The stimulation time should be met first, and therefore had the highest weight. In this way, when the simulation time constraint is not met, the objective gradient of the objective is highest for a change in simulation time. Once the simulation time constraint is met, the contribution of this objective is 0. Next, the optimization should ensure that the RMS deviation is below the maximum. Therefore, this objective has the second highest weight. When both constraints are met, their objectives do not contribute anymore. Then, the optimization will finally focus only on minimizing effort. Therefore, the weight was lowest for this objective. Note that the weight ratio between the three objectives should not affect the final result, and was chosen such that the optimization was successful in meeting the constraints.

Therefore, we created the following objective to achieve the desired behaviour:

(8) JSCONE(x(t),u(t))=J(u(t))

(9) +{100+10θRMSifθRMS>θRMS,max0otherwise

(10) +{1000(20−T)ifT<20s0otherwise.

During forward shooting, perturbations were added to the moment at the base of the pendulum. We ensured that the same perturbations were added each iteration by fixing the random seed. The perturbations were added as external moment to the pendulum each 0.1 s, starting at t = 0 s. The random moment was drawn from a uniform distribution with a given maximum amplitude, representing the task difficulty. We repeated each problem with 3 random seeds to account for variation due to the variability of the problem, and investigated the problem for which the optimal objective was lowest. All files used in SCONE, including the OpenSim model file, can be found in Koelewijn (2021).

Analysis

We performed analysis to investigate if co-contraction minimizes effort in systems with uncertainty, and to investigate the relationship between optimality of co-contraction and time delay. Table 2 summarizes the experiments and the parameters used for each. To show that co-contraction minimizes effort, it is sufficient to show one example where co-contraction minimizes effort. Therefore, we did not aim to replicate any specific biological situation and instead investigate a representative hypothetical problem. Since our problem was hypothetical, we have chosen different ranges of task precision, task difficulty, neural time delay, and activation time constants that illustrated the trends that we found.

Table 2 Overview of the experiments.

Parameters used in each of the experiments to investigate if co-contraction minimizes effort, and how this optimality related to time delay.

Experiment	Precision	Difficulty	Neural time delay	Activation time constant	
Effort landscape	5 deg	100 Nm	10 ms	10 ms	
Varying precision and difficulty	2, 3, 4, 5 deg	75, 100, 125, 150 Nm	0 ms	10 ms	
Neural time delay	5 deg	100 Nm	5, 10, 15, 20, 25 ms	10 ms	
Activation time	5 deg	100 Nm	0 ms	10, 30, 50, 70, 90 ms	

Effort landscape over different co-contraction levels

To investigate the effect of co-contraction on effort, we first visualized the effort landscape of feedforward control by solving optimizations with fixed levels of feedforward control. We use a maximum RMS deviation of 5 deg, a maximum perturbation amplitude of 100 Nm, and a time delay of 10 ms. Then, we solved optimizations with the feedforward control, u0, fixed between 0.02 and 0.22 with increments of 0.02 to investigate how the effort objective, J(u(t)) changes with the feedforward control. We also solved an optimization where the feedforward control is optimized, and compared this result to the solutions with fixed feedforward control.

A comparison of different tasks with varying precision and difficulty

To investigate the relationship between task precision and task difficulty, we solved optimizations for the controller with feedforward control for a range of tasks by varying the maximum RMS deviation (task precision) between 2 and 5 deg, with increments of 1 deg, and by varying the maximum perturbation amplitude (task difficulty) between 75 and 150 Nm, with increments of 25 Nm. Here, the neural time delay was equal to 0 ms. First, we discarded the tasks for which the combination of precision and difficulty could not be solved, meaning that the simulations did not last the full 20 s or the RMS deviation was larger than the maximum. Then, we selected the tasks for which the optimal feedforward control was larger than 0.01, the minimum activation, and investigated if the largest activation was close to 1. If this was the case, feedforward control is optimal not because of effort minimization, but it is required to perform the task, because the MTU strength is insufficient. Finally, for the tasks where the optimal feedforward control was larger than 0.01 and the maximum activation was not close to 1, we also solved optimizations for the controller without feedforward control. We compared these solutions to those with feedforward control for the tasks where both simulations lasted the full 20 s and where the RMS deviation was equal to or below the maximum. For those simulations, we compared the objective value, so the required effort, the feedback gains, and the co-contraction index (CCI) between the two simulations. We calculated the CCI as follows (Falconer & Winter, 1985):

(11) CCI=2(∫t1t2u1(t)dt+∫t2t3u2(t)dt)∫t1t3u1(t)+u2(t)dt,

where [t1, t2] is the time period where the activation in MTU 1 is lower than in MTU 2, and [t2, t3] denotes the time period where the stimulation in MTU 2 is lower than in MTU 1.

The effect of increased neural time delay

To investigate the relationship between time delay and co-contraction, we first investigated the effect of increased neural time delay. We used the task with maximum RMS deviation of 5 degrees, and a maximum perturbation amplitude of 100 Nm. We solved optimizations for both controllers with neural time delays between 5 ms and 25 ms, with increments of 5 ms. We investigated how the feedforward control developed with increasing neural time delay. Then we compared the solutions for the controller with and without feedforward control input using the objective, the CCI, and the controller gains as described before. Again, we only compared solutions for which the simulations lasted the full 20 s and the desired task precision was met. We also compared the optimal trajectories of the pendulum angle, muscle activation, muscle length, and muscle force for the controller with feedforward control to these optimal trajectories without feedforward control to investigate how these variables changed between both solutions.

The effect of a longer activation time

To investigate the relationship between time delay and co-contraction, we also investigated the effect of the muscle activation time using the same task as for the neural time delay. We solved optimizations for both controllers with activation time constants ranging between 0.01 and 0.09 with increments of 0.02, while setting the neural time delay to 0 ms. Our analysis was very similar to the neural time delay; we investigated how the feedforward control developed with increasing activation time constant. Then we again compared the solutions for the controller with and without feedforward control using the objective, CCI, and controller gains as described before, selecting only those solutions for which the simulations lasted the full 20 s and the desired task precision was achieved. We also compared the simulation outcomes between the two controllers, as well as to the simulation outcomes of the controller with neural time delay.

Results

Effort landscape over different co-contraction levels

We found that for the example task, the effort objective is not minimized without any feedforward input, but when the feedforward input is equal to just below 0.18 (Fig. 2). The effort landscape of the feedforward control seems quadratic, since the required effort increased both when more or less feedforward control was used. These results show that even when it is possible to have no co-contraction, it requires less effort to have feedforward control and thus co-contract both muscles.

Figure 2 Effort objective as a function of the feedforward input.

The blue dots show the optimal solutions found with prescribed feedforward input. The red star shows the optimal solution when feedforward input was optimized.

A comparison of different tasks with varying precision and difficulty

We found that co-contraction was optimal for a subset of the tested tasks with a sufficiently high precision, defined by the maximum RMS deviation, and difficulty, defined by the maximum perturbation amplitude (Table 3). Though all tasks could be simulated for the full simulation time of 20 s, it was not possible to meet the desired precision for all tasks. If this was the case, no optimal feedforward control was reported as the optimization was not considered successful. Furthermore, for the maximum perturbation amplitude of 150 Nm, we found that co-contraction was required because the maximum activation was reached (a > 0.995). However, for the four cells with bold font in Table 3, we found co-contraction was an optimal control strategy, with the maximum activation below 0.93.

Table 3 Optimal feedforward controls for tasks with different precision and difficulty.

Bold font indicate the tasks for which co-contraction was optimal without requiring maximum activation. A dash indicates that the desired task precision was not achieved, meaning that the constraint on the root mean square (RMS) deviation was not met.

Required precision (deg)	Maximum perturbation amplitude (Nm)	
	75	100	125	150	
2	0.010	0.056	–	–	
3	0.010	0.010	0.084	–	
4	0.010	0.010	0.046	0.210	
5	0.010	0.010	0.017	0.133	

We found that for two of the four highlighted tasks, it was possible to also use the controller with only feedback control, which required more effort, while for the other two tasks, the required precision was not achieved without feedforward control (Table 4). For the tasks with a maximum perturbation amplitude of 125 Nm, and a required precision of 4 or 5 degrees, we found that the objective was smaller for the optimal solution with co-contraction than for the optimal solution without co-contraction, while for the other two solutions, the actual RMS deviation was larger than the maximum, so the desired task precision was not achieved. When comparing the controllers with and without feedforward control, we find that the position and derivative gains of the optimal controller are smaller with feedforward control than without, and that this difference increases when the level of the optimal feedforward control increases. Furthermore, the CCI is about four times higher with feedforward control than without, and it increases with the level of the optimal feedforward control. However, the amount of co-contraction is still small, since the CCI was 2.8% for the results with co-contraction.

Table 4 Comparison of the RMS deviation, objective, feedback gains, and co-contraction index (CCI) of the optimal solutions with and without co-contraction for the highlighted tasks.

The objective (required effort) is given for all tasks that achieved the required precision. The position and derivative gain, as well as the CCI, are only reported for the tasks which achieved the required precision with and without feedforward control.

	Maximum perturbation amplitude	100	125	125	125	
	Required precision (deg)	2.0	3.0	4.0	5.0	
RMS deviation (deg)	No feedforward control	2.7	3.6	4.0	5.0	
	With feedforward control	2.0	3.0	4.0	5.0	
Objective	No feedforward control	–	–	0.1005	0.0893	
	With feedforward control	0.0839	0.1244	0.1001	0.0892	
Position gain	No feedforward control	–	–	4.113	2.975	
	With feedforward control	–	–	3.864	2.920	
Derivative gain	No feedforward control	–	–	0.251	0.220	
	With feedforward control	–	–	0.232	0.219	
CCI	No feedforward control	–	–	0.77%	0.98%	
	With feedforward control	–	–	2.84%	2.87%	

The effect of increased neural time delay

We found that the optimal feedforward control increased with an increasing neural time delay in the control (Fig. 3), and that the controller with feedforward control yielded lower objectives, and thus lower effort, than the controller without feedforward control (Table 5). The minimum feedforward control of 0.01 was optimal without neural time delay. The optimal feedforward control then increased in a somewhat linear fashion with the neural time delay, and co-contraction was optimal for all nonzero time delays. With a time delay of 10 ms or less, it was also possible to meet the required task precision without feedforward control (Table 5), and the objective was higher without feedforward control than with feedforward control. The difference in the objective with and without feedforward control increased for larger time delays, when optimal feedforward control was larger as well. Similar to the comparison of tasks without neural time delay (Table 4), we found that the optimal position and derivative gain were lower with feedforward control than without feedforward control. Furthermore, the CCI was consistently higher for the simulation with feedforward control, and increased to 29.9% for a time delay of 10 ms, while without feedforward control, the CCI remained close to 1%.

Figure 3 Optimal feedforward control as a function of neural time delay.

Table 5 Comparison of the RMS, objective, feedback gains, and CCI of the optimal solutions with and without co-contraction with different neural time delays.

The objective (required effort) is given for all tasks that achieved the required precision. The position and derivative gain, as well as the CCI, are only reported for the tasks which achieved the required precision with and without feedforward control.

	Time delay (ms)	0	5	10	15	20	25	
RMS deviation (deg)	No feedforward control	5.0	5.0	5.0	6.3	8.0	10.4	
	With feedforward control	5.0	5.0	5.0	5.0	5.0	5.0	
Objective	No feedforward control	0.0499	0.0670	0.120	–	–	–	
	With feedforward control	0.0499	0.0641	0.0957	0.134	0.171	0.207	
Position gain	No feedforward control	2.134	2.432	2.784	–	–	–	
	With feedforward control	2.134	2.098	1.730	–	–	–	
Derivative gain	No feedforward control	0.187	0.178	0.225	–	–	–	
	With feedforward control	0.187	0.160	0.145	–	–	–	
CCI	No feedforward control	1.36%	1.50%	1.19%	–	–	–	
	With feedforward control	1.36%	9.88%	29.9%	–	–	–	

A comparison of the joint angles, muscle activation, contractile element length, and muscle force during a simulation with and without co-contraction reveals small differences that indicate a different strategy is used by the controllers with and without feedforward control (Fig. 4). The pendulum angle in the simulation with feedforward control is slightly closer to zero (upright) than for the simulation without feedforward control (Fig. 4A, e.g. around 8 s). This strategy allows for larger deviations for large perturbations, e.g. at 9.8 s. Furthermore, in the simulation with feedforward control, the activation peaks are lower, while during periods with low activation, the activation is higher than in the simulation without feedforward control (Fig. 4B). The contractile element length is generally smaller for the simulation with feedforward control than for the simulation without feedforward control (Fig. 4C). Consequently, peak muscle forces are also lower in the simulation with feedforward control than in the simulation without feedforward control, but the difference between the peaks is smaller for the muscle force than for the activation. Furthermore, activation peaks, and therefore muscle force peaks are slightly delayed in the simulation with feedforward control compared to the simulation without feedforward control (Figs. 4B and 4D).

Figure 4 Comparison of optimization results with and without feedforward control, for a neural time delay of 10 ms.

Pendulum angle (A), activation (B), contractile element length (C), and muscle force (D) for the solution with feedforward control (blue) and without (red) are plotted for 5 s of the 20 s simulations. A solid line is used for muscle 1 (M1) and a dashed line for muscle 2 (M2) in (B)–(D).

The effect of a longer activation time

We found that the optimal feedforward control increased with an increasing activation time constant (Fig. 5), and that the controller with feedforward control led to lower objectives, and thus lower effort, than the controller without feedforward control. Similar to the neural time delay (Fig. 3), the optimal feedforward control increased somewhat linearly with the activation time constant. For the activation time constant of 0.03, it was possible to meet the desired task precision also without co-contraction. This solution had a higher objective (0.0934 vs. 0.0879) without feedforward control. The CCI was nonzero for both solution, and lower without feedforward control than with (0.94% vs. 14.1%).

Figure 5 Optimal feedforward control as a function of activation time constant.

Contrary to what was found for the neural time delay, we did not find a different strategy between both controllers when increasing the activation time constant, and we saw smaller differences in the pendulum angle, muscle activation, contractile element length, and muscle force when comparing controllers with different activation time constants than for the neural time delay (Fig. 6). The pendulum angle was very similar and the peak deviation at 9.8 s comparable between both controllers (Fig. 6A). Both controllers also yielded very similar contractile element lengths, though the controller with feedforward control generally had slightly smaller extremes than the controller without feedforward control (Fig. 6C). Similar to Fig. 4B, we observed smaller activation peaks and larger activation during periods where activation is low for the controller with feedforward control than for the controller without feedforward control (Fig. 6B), which led to similar small differences in muscle force (Fig. 6D), which again were smaller than observed in Fig. 4D.

Figure 6 Comparison of optimization results with and without feedforward control, for an activation time constant of 30 ms.

Pendulum angle (A), activation (B), contractile element length (C), and muscle force (D) for the solution with feedforward control (blue) and without (red) are plotted for 5 s of the 20 s simulations. A solid line is used for muscle 1 (M1) and a dashed line for muscle 2 (M2) in (B)–(D).

Discussion

We investigated the relationship between co-contraction and effort minimization, and found that for certain tasks in systems with uncertainty, co-contraction minimizes effort. This co-contraction is created by applying a non-zero feedforward control to an antagonistic MTU pair. To show this, we solved optimal control problems for different tasks using controllers with and without feedforward control on a one degree of freedom pendulum while minimizing effort. We found different reasons that yielded an optimal feedforward control larger than the minimum muscle activation of 0.01. In some cases, co-contraction was necessary because otherwise the MTUs where not strong enough to successfully perform the task. Then, the optimization was not able to find any feedback control gains to perform the task without co-contraction. However, in other cases, it was possible to also perform the task without feedforward control, but this required more effort. We also found that without feedforward control, the CCI was equal to around 1% for all tasks, meaning that the level of co-contraction is negligible. With feedforward control, it increased to 2.8% without time delay, to 14% with a larger activation constant, and to 30% with neural time delay for the tasks that could also be controlled without feedforward control. These results indicate that effort is minimized when an antagonistic muscle pair co-contracts, and that this co-contraction is especially optimal in muscles with time delay, either in the activation constant or through neural time delay. Therefore, we conclude that having feedforward control, and thus co-contraction, can minimize effort in environments with uncertainty, even when this co-contraction is not necessary.

We also investigated the relationship between time delay and co-contraction. To do so, we varied the activation time constant, and introduced a neural time delay in the control. For both, the amount of optimal feedforward control increased with an increase in time delay or time constant. In many cases, nonzero feedforward control was necessary to be able to solve the task. However, again, we found that certain tasks were solvable with and without feedforward control and that the combination of feedforward and feedback control, and thus antagonistic co-contraction, required less effort than only feedback control. We also found that with neural time delay, a different strategy was used with feedforward control than without, which changed the the optimal trajectory, while this change in strategy was not observed when the activation time constant was increased.

Our results show that co-contraction, contrary to what is often thought (Falconer & Winter, 1985; Winter, 2005), is not inefficient, and that it is not chosen out of necessity (Hogan, 1984; Berret & Jean, 2020), but also because it minimizes effort of movement in systems with uncertainty. Previous experimental work also showed already that uncertainty is taken into account when making movement decisions (Kim & Collins, 2015; Hiley & Yeadon, 2013; Donelan et al., 2004), and our results confirm this in simulation as well. De Luca & Mambrito (1987) previously showed that co-contraction was observed in environments with uncertainty, and our work explains this observation by showing that this co-contraction likely minimized muscular effort.

Our results have implications for predictive simulations of gait and other human movements, which are currently not sufficiently accurate for many applications. Instead of modeling dynamics deterministically, stochastic optimal control should be used to predict movements taking into account uncertainty. Currently, predictive simulations require a hand-crafted objective (Falisse et al., 2019) or a tracking term (Koelewijn & Van den Bogert, 2016) to be sufficiently accurate. It should be investigated if simulation accuracy might be improved by including uncertainty, without the aforementioned additional objectives or tracking term. For example, by taking into account uncertainty, a predictive gait simulation with a lower-leg prosthesis model could predict the co-contraction that is observed in experiments in the upper leg on the prosthesis side, which is currently not possible (Koelewijn & Van den Bogert, 2016). Then, predictive simulations could be used to improve prosthesis design, to find a design that is stable enough to not require co-contraction to minimize effort, because this co-contraction increases metabolic cost in gait of persons with a transtibial amputation (Waters & Mulroy, 1999).

Our results also highlight the care that should be taken when selecting parameters of the musculoskeletal system and the neural control algorithm. When we included neural time delay, a different trajectory was optimal for the controller with feedforward control than for the controller without feedforward control. This suggests that the choice of controller architecture could affect the results. Furthermore, we observed large differences in optimal feedforward control, and thus co-contraction level, when changing the activation time constant in a realistic range, since 10 ms is the default in OpenSim, while others use 35 ms (e.g. Lai et al., 2018). This suggests that optimization results are also highly dependent on the choice of musculoskeletal parameters.

Co-contraction of muscles is used as an indicator of impaired control (Hortobágyi & DeVita, 2000). However, our work shows that co-contraction does not necessarily indicate impaired function. Instead, co-contraction might be the most optimal control strategy for e.g. the elderly population, who have decreased strength and for whom falls could have dire consequences, such as fractures (Winter, 1995) or even death (Kannus et al., 1999). We showed that with decreasing maximum isometric force, thus decreasing strength, co-contraction becomes optimal, since it is not possible to perform the task otherwise. The task difficulty was represented by the maximum perturbation amplitude and by the task precision, because difficult tasks require one to remain close to the desired position, so to be more precise. By varying the task precision and difficulty, we also showed that the optimal and expected level of co-contraction increases with the task difficulty and precision. Elderly people might display more co-contraction due to the decrease in strength. Furthermore, they are more precise to avoid falls, and thus might aim to stay closer to the intended trajectory than younger people, which might further increase the amount of co-contraction.

We chose to use a simple model of uncertainty, by adding a perturbing moment to the base of the pendulum. However, in reality, uncertainty is more complex, and can be present internally or externally. Internal noise means uncertainty in the neural control, both in sensing (Bays & Wolpert, 2007) and stimulation, while external noise could be due to many sources, such as wind or uneven ground. We repeated the problem with internal noise added to the joint angle, to model sensory noise, or to the input, to model noise in stimulation. The solution was trivial with sensory noise, because the noise was removed from the system when the feedback gains were zero, such that no control at all is required. When input noise was added, it was again optimal to have non-zero feedforward control.

We also ensured that different muscle model parameters did not affect the results. We identified the maximum isometric force and the tendon slack length as main parameters that could influence the result. The maximum isometric force was set somewhat arbitrarily in combination with the task difficulty and precision. For the same difficulty and precision, a higher maximum isometric force would yield that overall muscle activation could be reduced, until eventually no co-contraction was required, while a lower maximum isometric force would increase the amount of co-contraction required, until eventually the task could not be solved anymore. Furthermore, we tested the effect of the tendon slack length. Currently, the tendon is slack at a smaller than optimal fiber length. Therefore, we tested the scenario where the tendon was exactly slack at optimal fiber length and the scenario where the tendon was already active at optimal fiber length by repeating one task for both scenarios. We found that the result was very similar for all three scenarios, meaning that the optimality of co-contraction was not due to the choice of tendon slack length.

We assumed symmetry in control, and therefore only optimized for one feedforward control, which was applied to both muscles. Alternatively, we could have optimized parameters for both muscles separately. However, the problem is entirely symmetric otherwise, which means that the control should also be symmetric, and any asymmetry in the control would be caused by the specific noise sample used in the simulation. Furthermore, during preliminary simulations we found that the feedforward control, as well as the position and derivative feedback gain converged to the same value for both muscles when optimized separately, while this approach required a longer simulation time. Therefore, we chose to simplify and speed up our pipeline and implement a single controller for both muscles.

System uncertainty was modelled with uniform noise to bound the maximum possible perturbation. If the perturbation moment was drawn from a normal distribution, it would be possible that the noise at a certain time instance is very large. Then, the optimal solution could have included co-contraction just to overcome this perturbation while still meeting the task constraint, while the muscles would not be strong enough otherwise, which would have affected our conclusion. Uniform noise is bounded and therefore its maximum is known.

Conclusion

In conclusion, we showed that co-contraction minimizes effort for certain tasks in uncertain environment, even when co-contraction is not necessary. This means that observations of co-contraction in human movements do not necessarily disqualify the minimal energy or effort theory. Furthermore, the optimal amount of co-contraction increases with the task difficulty and precision, as well as with the activation time constant and the neural time delay. We also found that for controllers with neural time delay, the optimal trajectory was dependent on the controller used, which means that care should be taken when designing controller architecture. Co-contraction is often thought of as inefficient and therefore avoided as much as possible. However, this work shows that co-contraction is not inefficient, but the combination of proactive and reactive control requires less effort than only reactive control. Therefore, training and rehabilitation should focus on removing the cause of co-contraction to increase movement efficiency, instead of removing co-contraction itself. Furthermore, optimal control problems of human gait should account for the nonlinearity of the human body and the system uncertainty to be able to create accurate movement simulations.

Additional Information and Declarations

Competing Interests

Author Contributions

Data Availability

Antonie van den Bogert is an Academic Editor for PeerJ.

Anne D. Koelewijn conceived and designed the experiments, performed the experiments, analyzed the data, prepared figures and/or tables, authored or reviewed drafts of the paper, and approved the final draft.

Antonie J. Van Den Bogert conceived and designed the experiments, authored or reviewed drafts of the paper, and approved the final draft.

The following information was supplied regarding data availability:

The files used to run optimizations in the software SCONE are available at GitHub: https://github.com/adkoele/MuscleCocontraction.

The code is available at Zenodo: Anne Koelewijn. (2021). adkoele/MuscleCocontraction: Code for Antagonistic Co-contraction Can Minimize Muscular Effort in Systems with Uncertainty (Published). Zenodo. https://doi.org/10.5281/zenodo.5549007.

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
