# Peer review of "Antagonistic co-contraction can minimize muscular effort in systems with uncertainty"

_PeerJ, doi:10.7717/peerj.13085_

## Round 0.1 · original submission · Major Revisions

This is an interesting paper that offers a new look into an old problem and it presents an impressive amount of work. The reviewers have made some comments that I believe will further improve the study and the readability of the paper.

Reviewer 1 ·

Excellent Review

This review has been rated excellent by staff (in the top 15% of reviews)
EDITOR COMMENT
This is in my opinion an excellent review, detailed and critical yet constructive, not in the least because the reviewer uses open questions without directing the authors in a specific direction. Comments are well motivated and presented with respect for the work that the authors have done.

Basic reporting

No comment

Experimental design

No comment

Validity of the findings

No comment.

Additional comments

This manuscript presents the results of a simulation study that demonstrates co-activation can be energy-optimal, particularly in noisy environments. This is a very important result, as it is often assumed that coactivation is sub-optimal. The authors' approach towards investigating whether co-contraction can be energy-optimal in noisy environments is well-thought-out, with several sub-studies included investigating the potential contributors to co-contraction being energy-optimal. Where the manuscript can be improved is the better motivate and describe some of the methodological choices, including the use of forward shooting methods and a simple inverted pendulum model.

Major Comments:
1) Although I agree that energy minimization is important for many of the behaviors we do on a day-to-day basis, the authors should be careful about overstating the importance of energy minimization for everything that we do (e.g., lines 46-47, lines 58-59, lines 291-294). Although tasks such as walking have likely evolved to be energy optimal, energy-optimality is not necessarily true of all tasks (e.g., novel task, sudden loss of balance, etc.). Moreover, while energy may be the main component being minimized in some tasks for healthy adults, this is not necessarily the case for impaired individuals who may be co-optimizing with other things (such as balance, pain, etc.). To be clear, I agree that energy minimization is important and that the results of this study demonstrate that co-activation can be energy optimal. But I advise the authors to tamper down some of the wording regarding energy minimization being the only thing that is important.

2) The motivation for the forward shooting approach combined with the simple inverted pendulum is confusing to me. Specifically:
a) The purpose of this paragraph starting on line 66 is to motivate the stochastic optimal control methods used in this study to effectively study the energy-optimality of co-contraction in noisy systems/environments. Several commonly used methods of optimal control are introduced, and their limitations are described. It is unclear as written if the linearized system approach mentioned in this paragraph would allow for the introduction of noise and it’s just the linearity that is “bad”, or if this method also suffers from being unable to include noise. [moderate/major]
b) The purpose of this paragraph starting on line 77 is to introduce the current methods for stochastic optimal control, including one recently developed by the authors using direct collocation. The authors’ direct collocation method overcomes the time-consuming limitations of previous approaches. However, the authors state that their new method does not scale well to models with a large number of degrees of freedom. Why is this? And why would forward shooting overcome this limitation? Could the authors somehow present a comparison of how large/difficult the problem becomes for each of these methods to motivate why forward shooting is the better approach? Finally, if the reason for shifting to forward shooting is because of the direct collocation approach becoming intractable when models have many degrees of freedom – why examine a single degree of freedom model in the current study?

3) One of the things I really like about this study is the different investigations towards showing co-activation is energy optimal for various reasons. However:
a) It is unclear by the end of the Introduction exactly what is being done in this study and how they related back to the hypothesis that co-contraction is energy optimal in a nonlinear system with uncertainty. The authors should consider re-writing this paragraph to more clearly describe the different “experiments” being done in this study, their predictions on what will happen in each of the experiments, and how these predictions would support the overall hypothesis.
b) There are a lot of different simulation “experiments” being run, but the purpose of each of them, how they are set up, etc. is hard to follow due to the organization of this section. The “Approach” section would be much easier to follow if it was split into sub-sections, similar to how the “Results” section is split up. For example, sub-headings could include: 1) Effect of co-contraction on effort, 2) Effect of task precision and co-activation, 3) Effect of neural time delay on co-contraction, 4) Effect of excitation-activation delay on co-contraction.
c) Related to point 3b above, there are also many different variables that are being tuned in each of the simulation “experiments”. It would be very helpful to have a table that summarizes for each “experiment: what variables are fixed and which ones are allowed to vary (and by how much).

4) Some methodological choices are unclear:
a) The authors state that they repeated each problem with 3 random seeds to account for variation due to the variability of the problem. What was done with the results of these 3 different repeats per problem? Were they averaged? Was the minimum of the three taken as the result? Or…?
b) It’s not clear why comparing the trajectories of the optimal simulations (lines 170-173) was only done for the neural time delay and muscle activation time investigations.
c) How were the weights of the objective function (Jscone) determined? And what effect might different weights have on the results of this study?
d) For the effect of adding neural time delay, why was the range 5-25ms chosen? For the effect of muscle activation delay, why was the range of 0.01-0.09 chosen? Were there any biological reasons for these choices? (line 166 + line 176)

5) The argument regarding potential changes in the elderly makes sense (Lines 317-319), but I would like to see it supported by data. What did deviation in the inverted pendulum look like with versus without the feedforward control? Did it deviate less, was the slope different, was the variability of movement different, etc.? In addition, Lines 331-334 regarding how max isometric force may affect co-contraction may also be a good argument for what might change in older adults and thus may be better suited to be included with the discussion about potential changes in the elderly.

Minor Comments:
- Introduction (lines 62-65): The introduction of the neural time delay as another reason for co-contraction is oddly worded and a little hard to follow. Perhaps the beginning of this sentence could be stated more clearly and better integrated with the next sentence.
- Methods (page 6): It would be helpful to more clearly denote in the manuscript that equation 1 is the “feedforward + feedback” controller and eq2 is the “feedback only” controller
- Methods (line 132): The motivation for why “noise” was added as perturbations at the base of the pendulum needs to be motivated
- Methods (line 157)”: “a perturbation amplitude of 100 Nm” means “a maximum perturbation amplitude of 100 Nm” right?
- Results: What is “experiment 1”, “experiment 2”, etc.? The authors should be consistent throughout the manuscript in how they refer to each set of optimizations.
- Discussion (lines 360-361): Towards the end of the Discussion section there is a sentence that states: “but the combination of proactive and reactive control requires less effort than only reactive control”. I really like this sentence. However, this may be the first time “proactive” and “reactive” are used together to describe feedforward and feedback control. These terms should be defined and used earlier.

Reviewer 2 ·

Basic reporting

no comment

Experimental design

no comment

Validity of the findings

no comment

Additional comments

The authors investigated the role of co-contraction when stabilizing one joint with an antagonist muscle pair in a simulation environment. It was found that when optimizing a cost function that penalizes movement effort, the model with a feedback controller also predicts co-contraction even it is not necessary for task achievement. My comments are listed below.

Major:
Eqs 8-10: Cost function is the sum of muscle activation, positional error and success. This is the authors’ definition of “effort”. The ratio parameters were determined quite arbitrarily (100, 10, 100 in Eqs. 9&10). Since three components of the cost function have different units with different ratios, it is difficult to interpret the balance among the three. How are these ratios determined and how optimization results depend on them?

In current design (Eq.1), co-contraction simply makes the plant stiff at its current position – even when this position deviates from the neutral position. So, I doubt if co-contraction can contribute to stabilization around the neutral position. An alternative design could be that the co-contraction always works around a certain equilibrium position (e.g., the neutral position in the case of this study) which the plant is attracted to.

The two feedback gains were also optimized. The author would need a comparison between with and without feedforward control when the two gains for feedback control are the same: first find the optimal feedback gains for the model with pure feedback control. Then add the feedforward controller and run the optimization with the same feedback gains. Would the cost be smaller in this case?

Neural delay has a maximal value of 25ms. It is not clear how this could be related to real physiological system. The values are within the range of fast feedback loops in the neural system (i.e., spinal level). Does the result have any implication about higher-level feedback control (with neural delay > 50ms)?

Minor:
Eq. 2, why was the feedforward control set to 0.01 instead of 0? E.g., in Fig.2 the feedforward signal starts at 0.

Line 136: Was there a particular reason why the authors did not simulate the gravity? Gravitational torque can cause a fall when exceeding muscle torques.

What is the rational to investigate different time delays and activation constant? Would this also imply some kinds of impaired control in patients with movement disorders?

Line 104: It is better to use effort than energy, to be consistent with the text.

Lines 261-262: Would larger feedback gains be sufficient to perform the task without co-contraction, even when MTU were (also a typo in text) not strong enough?

---

## Round 0.2 · Minor Revisions

Dear Anne,

One of the reviewers has made a final minor suggestion that I hope you can incorporate. The suggestions is as follows: Eqs. 9 and 10: please also indicate ''...=0, if otherwise''. Congratulations, your paper will be acceptable for publication after this.

best regards
Jaap van Dieën

Reviewer 1 ·

Basic reporting

No comment

Experimental design

No comment

Validity of the findings

No comment

Additional comments

The authors have sufficiently responded to my prior critiques.

Reviewer 2 ·

Basic reporting

no comment

Experimental design

no comment

Validity of the findings

no comment

Additional comments

The authors have addressed my comments in a satisfactory way and I believe that the manuscript has been much improved. I have only one further suggestion:
Eqs. 9 and 10: please also indicate ''...=0, if otherwise''.

---

## Round 0.3 · accepted · Accept

beste Anne en Ton,

Gefeliciteerd. Dit paper is nu geaccepteerd voor publicatie en het is mijns inziens een waardevolle bijdrage aan de literatuur.

groet
Jaap